# Sequential Extraction of Proanthocyanidin Fractions from *Ficus* Species and Their Effects on Rumen Enzyme Activities In Vitro

**DOI:** 10.3390/molecules27165153

**Published:** 2022-08-12

**Authors:** Pushpendra Koli, Sultan Singh, Brijesh K. Bhadoria, Manjree Agarwal, Suman Lata, Yonglin Ren

**Affiliations:** 1Indian Council of Agricultural Research (ICAR)-Indian Grassland and Fodder Research Institute, Jhansi 284003, UP, India; 2College of Science, Health, Engineering and Education, Murdoch University, South Street, Murdoch, WA 6150, Australia; 3Chem Centre, Resources and Chemistry Precinct, Bentley, WA 6102, Australia

**Keywords:** proanthocyanidins, flavonoids, *ficus religiosa*, ficus racemosa, ruminal enzymatic activity and rubisco

## Abstract

Three proanthocyanidin fractions per species were sequentially extracted by 50% (*v*/*v*) methanol–water, 70% (*v*/*v*) acetone–water, and distilled water from leaves of *Ficus racemosa* (fractions FR) and *F. religiosa* (fractions FRL) to yield fractions FR-50, FR-70, FR-DW, FRL-50, FRL-70, and FRL-DW. Fractions were examined for their molecular structure, effect on ruminal enzyme activities, and principal leaf protein (Rubisco) solubilization in vitro. All fractions except FRL-70 contained flavonoids including (+) catechin, (−) epicatechin, (+) gallocatechin, (−) epigallocatechin, and their -4-phloroglucinol adducts. The fractions FRL-50 and FRL-DW significantly (*p* < 0.05) inhibited the activity of ruminal glutamic oxaloacetic transaminase and glutamic pyruvic transaminase. All fractions inhibited glutamate dehydrogenase activity (*p* < 0.05) with increasing concentration, while protease activity decreased 15–18% with increasing concentrations. Fractions FRL-50 and FRL-DW completely inhibited the activity of cellulase enzymes. Solubilization of Rubisco was higher in *F. religiosa* (22.36 ± 1.24%) and *F. racemosa* (17.26 ± 0.61%) than that of wheat straw (WS) (8.95 ± 0.95%) and berseem hay (BH) (3.04 ± 0.08%). A significant (*p* < 0.05) increase in protein solubilization was observed when WS and BH were supplemented with FR and FRL leaves at different proportions. The efficiency of microbial protein was significantly (*p* < 0.05) greater in diets consisting of WS and BH with supplementation of *F. racemosa* leaves in comparison to those supplemented with *F. religiosa* leaves. The overall conclusion is that the fractions extracted from *F. religiosa* showed greater inhibitory effects on rumen enzymes and recorded higher protein solubilization in comparison to the *F. racemosa*. Thus, PAs from *F. religiosa* are potential candidates to manipulate rumen enzymes activities for efficient utilization of protein and fiber in ruminants.

## 1. Introduction

Tree foliages are a source of protein, energy, and minerals for many herbivorous animals [1]. Such foliage may contain anti-nutritional factors such as polyphenols, tannins, alkaloids, and proanthocyanidins, which limit their utilization [2,3]. Some species of fodder trees are regarded as valuable components of agro-silvicultural systems and have been maintained for centuries as feed for small ruminants. *Ficus religiosa* and *F. racemosa* are used for this purpose in subtropical agro-silvicultural systems in central India.

Crude protein (10.7–11.5%), neutral detergent fiber (42.5–49.3%), lignin (5.6–10.8%), and in vitro dry matter digestibility (37.9–55.8%) of leaves of these *Ficus* species were assessed in our previous study [4]. Proanthocyanidins (PAs) are complex bioactive compounds that are difficult to characterize and quantify because of their complex molecular structure. PAs are categorized based on monomer unit linkage, which consists of either flavan-3-ol or flavan-3–4 diol or leucoanthocyanidin [5]. PAs are oligomeric and polymeric flavan-3-ols, and their sizes differ in their degrees of polymerization [6]. These vary tremendously in molecular weight and structure. Their multiple phenolic hydroxyl groups lead to the formation of complexes primarily with protein and, to a lesser extent, with metal ions, amino acids, and polysaccharides. PAs are considered to have both adverse and beneficial effects depending on their concentration and chemical nature besides other factors such as the animal species consuming it, the physiological state of the animal, and the percentage composition of the diet [4].

The use and selection of appropriate solvent is equally important to obtain plant extracts. The amount and types of solvent are to be considered important parameters while extracting tannin-related compounds [7]. However, greater extraction efficiency and higher performance of solvent extraction can be achieved by using green technologies such as supercritical CO_2_ extraction method and robotic solvent automation [8,9], whereas the most common separation techniques for PAs are column chromatography, paper chromatography, thin layer chromatography (TLC), and high-performance liquid chromatography (HPLC) [10]. UV analysis is also a prominent technique [11]. However, TLC and paper chromatography remain the most versatile methods for the detection and separation of phenolics from crude extracts. Spray reagents for detection of phenolics were reviewed by Markham [12]. Recent advances have shown that high performance thin layer chromatography (HPTLC) can provided better separation of complex mixtures than can TLC [13].

Studies on screening plant extracts have become an important research area for researchers globally to develop alternative feed additive to manipulate rumen ecology for reducing greenhouse gases [14]. In animal studies, consumption of PAs has been reported to protect against peroxynitrite-induced endothelial dysfunction and chemically induced cancer [15]. Being natural products, plant extracts are attractive as rumen modifiers, which are environmentally friendly and safe in food production systems [16]. Oil palm leaf extract (OPLE) significantly (*p* < 05) reduced CH_4_ production and total methanogens and methanobacteriaceae [17]. Tannins influence protein digestion and fermentation; non-tannin phenolics (NTP) including all flavonoids decrease the in vitro CH_4_ production without impairing in vitro organic matter digestibility (IVOMD) [18]. The ability of NTP to reduce CH_4_ is related to their chemical structure as phenols with two or more hydroxyl groups seemed to have higher efficiency than those containing only one. Structures diversity of CTs affects rumen in vitro CH_4_ production in sainfoin accessions. Procyanidin: Prodelphinidin (PD: PC) ratio is important and negatively associated with CH_4_ production [19]. Globally, substantial information is available on the nutritive value of tree foliage in respect to protein, minerals, digestibility, intake, level of supplementation, enteric methane mitigation, etc. [20,21]. However, limited phytochemical properties are explored for animal studies in the Indian context [22]. In addition, there is a lack of knowledge on the PAs, which play an important role in animal health [23].

In this study, the objective was to isolate and characterize PAs in a sequential extraction with series of solvents based on deferent polarity from tree foliage of *F. religiosa* and *F. racemosa* and to assess their effect on in vitro ruminal enzyme activities for better utilization in animal nutrition.

## 2. Results and Discussion

### 2.1. Extraction and Molecular Characterization of Proanthocyanidins

Proanthocyanidins or condensed tannins are widely distributed within plants, particularly woody plants. These are mainly composed of (+) catechin or (−) epicatechin units, and these units attribute the properties to the condensed tannins. The basic structural unit of these phenolic polymers is flavan-3ol. All isolated fractions contained phenolic compounds except FRL-70 (eluted by 70% (*v*/*v*) acetone–water). The other fractions, FRL-50 (eluted by 50% (*v*/*v*) methanol–water), FRL-DW (eluted by distilled water), FR-70 (eluted by 70% (*v*/*v*) acetone–water), FR-50 (eluted by 50% (*v*/*v*) methanol–water), and FR-DW (eluted by distilled water), showed the presence of (+) catechin (I), (−) epicatechin (II), (+) gallocatechin (III), (−) epigallocatechin (IV) along with their -4-phloroglucinol adducts (Figure 1). The (+) catechin was not detected in FRW-DW and FR-50. The non-detection of certain flavonoids is due to poor solvent extraction efficiency when levels of organic solvents were low. Methanol and acetone are the most effective solvents for extraction of most flavonoids [24]. In the present study, 50% methanol and 70% acetone were the most effective solvents for extraction of PAs from both *Ficus* species. The acid catalyzed cleavage of phloroglucinol adduct of different proanthocyanidins and monomeric flavan-3-ols was examined in HPLC chromatograms measured at 280 nm with the following standards used as references and eluted in the order of: (+) gallocatechin, (−) epigallocatechin, (+) catechin, (−) epicatechin, and a similar order was followed by -4-phloroglucinol adducts with retention times (Rt) of 15.31–48.71 min. Similar trends for above compounds were observed in *Vicia faba* [25].

The highest (0.524 ± 0.03) and lowest (0.098 ± 0.01) catechin was found in FR-DW and FR-70, respectively, whereas it was not detected in FRL-DW and FR-50 (Table 1). A similar trend was observed for epicatechin. Gallocatechin and epigallocatechin was present at the greatest levels in FR-70 (0.458 ± 0.13) and FR-DW (0.315 ± 0.11), and lowest in FR-DW (0.035 ± 0.02) and FR-50 (0.016 ± 0.07). Similar results were reported in *Litchi chinensis* [26], *Saracaasoca* [27] and *Ephedra sinica* [28]. HPLC analysis of acid-catalyzed cleavage of the phloroglucinol adduct of different proanthocyanidins revealed flavan-3-ol in monomeric and adduct. All compounds varied in their polymeric structure and the degree of polymerization in the monomer and phloroglucinol adduct ratio. The observed ratios were recorded as: FRL-50, 3.06; FRL-DW, 3.99; FR-70, 5.09; FR-50, 3.46, and FR-DW, 2.63. The degree of polymerization helps to enhance the stability of metal-flavonoid complexes, and these initial metal-flavonoid complexes are very good cytoprotectives against superoxide radical scavenging [29]. Furthermore, the adduct represented the extender unit (EU), whereas the terminal unit (TU) of the molecule was represented by monomers corresponding to the 4 β linkage in the molecule. FRL-50 consisted of 24% TU and 76% EU. FRL-DW had 20% TU and 80% EU (Appendix A). Proanthocyanidins FR-70, FR-50, and FR-DW possessed EU 84, 78, and 68% and TU 16, 22, and 32%, respectively.

Flavanols are present naturally in many forms, from monomeric to polymeric, and they differ from each other structurally in the configuration of carbon C-2, the hydroxylation pattern of the rings, the type of linkage between each unit, and the degree of galloylation [30]. The common configuration in the monomers of the flavanol is 2*R*:3*S* or 2*R*:3*R* in episomer and enantiomers such as ent-epicatechin (2*S*:3*S*). In Appendix A and Figure 1, (+) Catechin and (−) epicatechin have hydroxylation positions at 3,5,7,3′,4′ carbon number in the flavan-3-ol unit but differed in R-S configuration: 2*R*:3*S* and 2*R*:3*R,* respectively. In (+) gallocatechin and (−) epigallocatechin, one additional hydroxyl group attached at the 5′ position. The subunits of anthocyanin in *F*. *racemosa* and *F. religiosa* were predominated by cyanidin subunits with delphinidin in a ratio of 50:50 (Table 1). The variation in subunits of proanthocyanidins varied from cultivar to cultivar [31]. Thus, different amounts of anthocyanin were found in the two tree species examined in this study.

### 2.2. Effect of Proanthocyanidins on Ruminal Enzymatic Activities

The activities of ruminal glutamic oxaloacetic transaminase (R-GOT), glutamic pyruvic transaminase (R-GPT), and cellulase are shown (Table 2). All the fractions along with tannic acid (TA) and gallic acid (GA) reduced rumen enzyme activities significantly (*p* < 0.05). There was a liner decrease in the activity of R-GPT and R-GOT with increase in their concentrations. The EC_50_ of TA for R-GOT was 20.54 ± 1.13 mg/mL and 17.54 ± 0.13 mg/mL for R-GOT of the protozoan and bacterial fractions, respectively. GPT activity also decreased (*p* < 0.05) with increasing concentration of TA, and its EC_50_ was 17.52 ± 0.70 mg/mL and 17.12 ± 0.15 mg/mL for protozoan and bacterial fractions, respectively. GA also lowered R-GOT and R-GPT in both in protozoan and bacterial fractions. The effect on R-GPT activity was nearly equal to that of TA for protozoan and bacterial fractions, whereas the GA was more significantly (*p* < 0.05) effective to R-GPT in the protozoan fraction (EC_50_ 6.89 ± 0.12 mg/mL) than that of the bacterial fraction (EC_50_ 15.52 ± 0.53 mg/mL). The FRL-50 and FRL-DW both (*p* < 0.05) inhibited R-GOT and R-GPT activities. FRL-50 had greater inhibition of R-GOT (protozoan fraction) activity than R-GOT (bacterial fraction) with EC_50_ 4.72 ± 0.42 mg/mL and 7.12 ± 0.02 mg/mL, respectively. The EC_50_ of FRL-DW was 10.67 ± 1.12 mg/mL for R-GOT (protozoal) and 7.69 ± 1.25 mg/mL for R-GOT (bacterial), respectively. In the case of R-GPT activity, FRL-50 was more potent as evidenced from lower EC_50_ values of 5.44 ± 0.49 mg/mL and 5.28 ± 0.63 mg/mL for protozoal and bacterial fractions, respectively. The FRL-DW had a higher EC_50_ to inhibit R-GPT (13.59 ± 1.42 mg/mL and 16.69 ± 1.44 mg/mL) for protozoal and bacterial fractions, respectively. On the other hand, FR-70 and FR-50, polyphenolics from *F. racemosa,* inhibited R-GOT (protozoal) more effectively with an EC_50_ 17.17 ± 1.01 and 12.33 ± 1.26 mg/mL, respectively, but they were less active against the R-GOT bacterial fraction with relatively higher EC_50_ of 19.17 ± 1.00 and 16.33 ± 0.50 mg/mL, respectively. FR-50 exhibited significantly (*p* < 0.05) higher inhibition/reduction in GPT activity than FR-70 in both protozoal and bacterial fractions. The inhibition of R-cellulase activity was affected by the presence of phenolics in the rumen liquor (Table 2). The cellulase enzyme in rumen liquor was more sensitive to GA in comparison to TA as the effective concentration to inhibit 50% activity (EC_50_) was 6.62 ± 0.56 mg/mL and 124.32 ± 1.63 mg/mL for GA and TA, respectively. The activity of ruminal cellulase was completely inhibited by the polyphenolics from *F. religiosa*. FR-50 showed greater potential (*p* < 0.05) to inhibit 50% of cellulase activity at 9.70 ± 0.54 mg/mL whereas FR-70 possessed an EC_50_ of 24.72 ± 1.25 mg/mL.

The decrease in R-GOT, R-GPT, and R-cellulase in the presence of proanthocyanidins fractions could be related to the antimicrobial nature of these compounds or their metabolites produced during the fermentation process. Our studies revealed that presence of catechin, epicatechin, and other derivatives can reduce enzymatic activity. Similar results were shown in a study of male cow rumen fermentation by Oskoueian et al. [32]. In the fraction FRL-50, all components of proanthocyanins were present in abundance and the inhibition of all enzymes occurred. The mode of action of proanthocyanins is not exactly known but they have direct and indirect effects on cellulose digestion. The inhibitory effect of proanthocyanidins on cellulose digestion was also observed in legume forages by McAllister et al. [33].

The presence of phenolics in the substrate reduced the oxidation rate of NADPH and NAD, resulting in reduced R-GDH activity in protozoal and bacterial fractions of the rumen liquor. The linear decrease in R-GDH activity was dependent of the quantity of TA and GA. The effect of TA on R-GDH activity was more pronounced on the protozoal fraction than in the bacterial fraction. There was decline of 12 units on the protozoal fraction, while in the bacterial fraction it declined by 7 units (Figure 2). On the contrary, the reducing effect of GA was more pronounced on bacterial (9 units) than on protozoal fractions (4 units). FRL-50 was able to reduce the R-GDH (protozoa) to the extent 0.321 IU/L and complete inhibition of R-GDH at each concentration. This effect may be interpreted as FRL-50 being toxic to R-GDH from the bacterial part of rumen liquor in the same way FRL-DW was able to stop the activity of R-GDH at a concentration of 14 mg/mL. However, the activity of the protozoal fraction was also lowered with the increase of concentration of FRL-DW. FR-50 more potently inhibited R-GDH protozoal fraction activity. It completely inhibited R-GDH activity at 10 mg/mL and R-GDH (bacterial fraction) at 14 mg/mL. However, both lowered the GDH activity with increasing concentration. The effect on protease activity of all fractions was examined at concentrations from 2–10 mg/mL (Figure 3). The R-protease activity in every fraction of FR and FRL declined with increasing concentration and time of incubation. FRL-DW exhibited 15–18% decrease in protease activity at each concentration. In the case of FRL-50, the decline in protease activity was 18–20%, as with FRL-DW. In the case of *F. racemosa,* there was a decrease of 20–30% and 18–20% in protease activity by FR-50 and FR-70 fractions, respectively. FR-DW was shown to decrease protease activity by 25–60%. Earlier studies conducted on bark extracts from different medical plants showed that isolated polycyclic phenolics inhibited protease activity from 22–56% [34].

The interaction between PAs and proteolytic enzymes may inhibit proteolysis due to steric interference with the binding of the protease with susceptible sites on dietary protein. The degree of binding is dependent on the types of protein and types of PAs. The exact mode of action is not fully known but the reduction in proteolysis may be due to direct effects of PAs on microbial proteolytic enzymes or indirect effects on rumen metabolite concentration, which can inhibit proteolysis in some bacteria [35]. In line with our finding, Jones [36] reported that *Onobrychis vicifolia* CTs had profound effects on growth of five rumen bacteria. At the higher tannin protein ratio, inhibition in proteolysis occurs due to polyphenolic compounds adhering to the protein surfaces to cause interference with the interaction of the enzyme substrate [37]. Proanthocyanidins were considered beneficial if they could bind to the protein [38], and those that did not bind were considered harmful because they lowered the ruminal digestion, especially hemicellulose [39].

### 2.3. Effect on Principal Protein (RUBISCO) Solubilization and Microbial Protein Efficiency

Maximum solubilization of Rubisco in FRL (22.36 ± 1.24%) was followed by FR (17.26 ± 0.61%) (Table 3). Solubilization was comparatively higher (*p* < 0.05) than that of roughage WS (8.95 ± 0.95%) and berseem hay (3.04 ± 0.08%). Addition of FR and FRL to WS increased dietary protein solubilization to 16.61 ± 1.04% and 22.47 ± 1.37% at a ratio of 1:1, respectively. In case of FR, the solubilization again increased at 2:1 (28.58 ± 1.48) and then declined at 3:1 (15.29 ± 0.61). At 2:1 and 3:1 ratio of FRL, WS Rubisco solubilization declined to 22.47 ± 1.37% and 5.59 ± 0.34, respectively. This indicated a regulation in protein solubilization with certain quantities of PAs in the WS. Further supplementation of FRL leaves to BH in 1:1, 2:1, and 3:1 proportion gradually increased dietary protein to 14.47 ± 0.74, 17.80 ± 1.26%, and 25.90 ± 1.18%, respectively. For FR in addition to BH, the protein solubilization increased from 18.35 ± 1.48 at 1:1 to 21.70 ± 3.00% at 2:1 proportion and lowered at a 3:1 ratio (11.81 ± 1.01%). The determination of NRNAQ in rumen liquor after in vitro dietary protein solubilization from leaves and their addition with roughage WS and BH in combination of 1:1, 2:1, and 3:1 revealed that among the top feeds along with *FR,* it synthesized more microbial protein as evident by NRNAQ 0.22 ± 0.01 than *FRL* (0.21 ± 0.02). Indeed, the microbial synthesis of FR and FRL was higher (*p* < 0.05) to sole WS and BH as their NRNAQ was 0.16 ± 0.01 and 0.17 ± 0.01, respectively. The NRNAQ was low in FRL (0.19 ± 0.01) and FR (0.15 ± 0.01) at 1:1 with WS. Supplementation of top feed leaves to BH in 1:1 ratio showed a considerable increase in microbial protein synthesis relative to sole WS or BH as evidenced by NRNAQ 0.22 ± 0.01 and 0.26 ± 0.03 with FRL and FR, respectively. Increasing the level of top feed leaves in a ratio of 2:1 with WS the NRNAQ did not alter much. However, with FRL there was a slight decrease in NRNAQ (0.16 ± 0.01), whereas addition of FRL and FR to BH in a ratio of 2:1 lowered microbial protein synthesis to 0.17 ± 0.01 and 0.21 ± 0.01, respectively. Increase of top feeds in 3:1 ratio with WS declined NRNAQ as 0.18 ± 0.01 and 0.19 ± 0.01 and with berseem hay the NRNAQ was 0.17 ± 0.0 and 0.21 ± 0.02 for FRL and FR, respectively.

The digestion of forage in the rumen occurs via a combination of solubilization and degradation. Solubilization is an important action and comprises the release of protein from the cell wall into ruminal fluid with chewing in the mouth. On the other hand, degradation is a catabolic process facilitated by microbial proteases resulting in peptides, amino acids, and ammonia [40]. These two processes are independent of one another. Therefore, solubilization resembles the loss of principal protein (Rubisco) from the feed substrate during in vitro incubation under ruminal conditions, and degradation could be determined by identifying the changes in proteins. In green fodder, chloroplasts accounted for 75 to 80% of total leaf protein [41] as Ribulose-1, 5 biphosphate carboxylase-oxygenase and it is the major soluble protein. It catalyzes the first step in the Calvin reductive pentose phosphate cycle. Presence of PAs increases duodenal NAN, flow per unit total N eaten, and reduces ammonia concentration in the rumen [42]. Therefore, loss of Rubisco from tree leaves and their combination with WS and BH during in vitro dry matter disappearance revealed how the presence of different PAs behave in solubilization. An increase in protein from *Trifolium repens* and *Lotus corniculatus* solubilization with addition of phenolics (condensed tannins) has been reported [40]. We observed that addition of FR and FRL tree leaves to WS considerably increased Rubisco solubilization. Molecular structure and molecular weight of PAs influences the microbial protein efficiency. There are reports of both increase and decrease in production of microbial proteins upon addition of phenolics. Bento et al. [43] reported the reduction in microbial protein by Mimosa tannin, whereas an increase in microbial protein synthesis efficiency was observed by Gatachew et al. [44] with supplementation of tannin in alfalfa (*Medicago sativa*) hay. Similarly, in our findings, the impact of addition of *Ficus* leaves to wheat straw and berseem hay increased microbial protein synthesis efficiency.

## 3. Materials and Methods

### 3.1. Reagents

Catechin, epicatechin, Gallocatechin, epigallocatechin, catechin-4-phloroglucinol, epicatechin-4-phloroglucinol, gallocatechin-4-phloroglucinol, epigallocatechin-4-phloroglucinol, and Sephadex LH-20 were purchased from Sigma, USA. All other reagents and solvents used were of analytical grade. UV spectra were measured on a UNCAM UV/vis spectrophotometer (Newington, CT, USA). TLC, column chromatography, and paper chromatography were performed on precoated Si GF^256^, Si gel (60–120 Mesh, Merck, New Delhi, India).

### 3.2. Plant Materials

The leaf samples from mature trees of *F. religiosa* and *F. Racemosa* were collected from the Central Research Farm of ICAR-Indian Grassland and Fodder Research Institute and ICAR-Central Agro-Forestry Research Institute, Jhansi. These tree leaves are often used by the farmers for animal feeding. Collected samples were dried under shade on the cemented floor of the laboratory followed by drying in a hot air oven at 60 °C until the constant dry weight was achieved. Dried samples were powdered to pass through 1 mm sieve using electrical Willey mill. Powdered samples were stored in plastic containers (Tarson make) for further chemical analysis.

### 3.3. Extraction and Molecular Characterization of Proanthocyanidins

Tree leaves of *Ficus religiosa* (FRL) and *F. racemosa* (FR) (2 kg of each) were extracted with 70% acetone containing 0.1% ascorbic acid. The solvent was removed under vacuum in a rotatory evaporator at 40 °C. The aqueous phase was washed subsequently with chloroform, diethyl ether, and ethyl acetate. The remaining aqueous extract was diluted with an equal amount of methanol (MeOH) and charged over a pre-equilibrated Sephadex LH-20 (30 × 2.5 cm) column. (Appendix A, Appendix A).

All extracted compounds were subjected to qualitative phytochemical investigations for presence of phenolics and flavonoids in compounds through Shinoda, vanillin/HCL, and FeCl_3_ tests along with Thin Layer Chromatography (TLC) and paper chromatography (PC) profiling then tested against enzymes (Figure 4). To determine the monomeric unit, each isolated proanthocyanidin was independently treated with phloroglucinol in the presence of 1% HCl in alcohol for 4 h. After removal of the solvent, the product was diluted with water and extracted with ethyl acetate followed by its evaporation to dryness and dissolved in 80% MeOH quantitatively for HPTLC (cellulose), solvent system (TBA, and 6% acetic acid), and HPLC (equipped with UV/vis detector and RP ODS column (25 cm × 4 mm, id) at ambient temperature with a solvent system of acetic acid (1%) and methanol at 1 mL/min. The retention time (Rt) in HPLC varied from 15.31 to 48.71 min for the above flavonoids. The quantification of flavonoids was done using the regression equation y = mx + c, which was obtained from the calibration curve generated using a standard solution at different dilutions. The limits of detection and quantification were determined at signal-to-noise (S/N) ratios of 3 and 10, respectively. To study the subunits, each isolate was hydrolyzed completely with 5% HCl in n-Butanol for 1.5 h in a water bath. The resultant product after removal of acid was examined chromatographically on Whatman No. 1 and Whatman No. 3 paper and a cellulose TLC plate using Forestal as the solvent system for their resolution. The resolved spots were identified by co-comparison of R_f_ values of pure samples obtained from Sigma Aldrich USA. Estimates of anthocyanidin were acquired using UV spectroscopy.

### 3.4. Ruminal Enzymatic Activities

For estimating the activity of extra-cellular enzyme (cellulase), the technique described by Mandel and Weber (1969) was followed [45]. For intra-cellular enzymes (GPT: Glutamic pyruvic transaminase; EC 2.6.1.2, GOT: Glutamic oxaloacetic transaminase; EC 2.6.1.1, and GDH: Glutamate dehydrogenase; EC 1.4.1.2), rumen liquor samples of sheep were centrifuged at 2500 rpm at 4 °C for 5 min to separate protozoal and bacterial fractions as residue and as supernatant, respectively. From these fractions, bacteria and protozoa-rich enzyme extracts were prepared in 0.1 M phosphate buffer of pH 6.8 [46]. The activities of the GOT and GPT in both microbial fractions were assayed following the method of Yatzidis [47]. The method by Strecker [48] was followed to measure the specific activity of the GDH enzyme. The oxidation of the glutamate was measured in both microbial fractions (protozoal and bacterial) by the decrease in optical density by measuring the rate of change in absorbance at 340 nm caused by reduction of di-phospho pyridine nucleotide (DPNH). Protein in the enzyme extracts was estimated by the method of Lowry [49] to work out the specific activities of the assayed enzymes. Proteolytic activity of the bacteria was studied by estimating undigested protein from casein as described by Blackburn and Hobson [50] with a modification used by Kumar and Singh [51].

### 3.5. Principal Protein (RUBISCO) Solubilization and Microbial Protein Efficiency

Five hundred mg of green lyophilized and powdered tree leaves of *F. racemosa* and *F. religiosa* and their combinations of 1:1, 2:1, and 3:1 with wheat straw (WS) and Berseem hay (BH) separately were incubated at 37 °C with 40 mL of strained rumen liquor from grazing sheep and 10 mL of CO_2_ saturated buffer (containing mM 117 NaHCO_3_, 26Na_2_HPO_4_, 8NaCl, 8KCl, 0.2CaCl_2_, and 0.3MgCl_2_) for 48 h in a full anaerobic environment and filtered through a G-1 crucible. The filtrate was centrifuged at 500× *g* for 15 min to remove the feed particles. The supernatant was then centrifuged at 20,000× *g* for 30 min at 4 °C. This process was repeated three times. The pellet was washed with saline and preserved for determination of microbial protein and RNA after lyophilization. The residue collected in the crucible was used for Rubisco determination by the Bradford method [52] after protein extraction with protein buffer and SDS-PAGE electrophoresis with ILQ software. To determine the microbial protein efficiency the prepared pellet from filtrate was used for N determination by the Kejldhal method [53] and microbial purine as RNA by the method of Zinn and Owen [54]. The efficiency of microbial protein was calculated as the microbial nitrogen:microbial RNA Quotient (NRNAQ).

### 3.6. Data Analysis

Microsoft Excel 2015 and R were used for statistical analysis. Analysis of variance (ANOVA) was performed in R to evaluate the measurements of enzymatic activities and means are considered as significantly different at *p* < 0.05 by post-hoc analysis with Tukey’s test.

## 4. Conclusions

Six fractions of PAs were extracted and quantified from two *Ficus* tree species. The results indicated that isolated PAs inhibited the activities of rumen enzymes (R-GOT, R-GPT, and R-cellulase) and protein degradation in vitro. The mode of action of PAs is not fully known, but results revealed the relationship between proanthocynidins’ chemical structure and rumen enzymes’ activities, which may be utilized to manipulate the rumen fermentation, particularly the fiber (cellulose) and protein utilization (transaminases—GOT and GPT, deaminase—GDH). Further studies (in vivo) are required to better understand the relationship of the molecular structure of PAs with the ruminal enzymatic activities and microbial protein metabolism. In the future, the PAs rich plant extracts could be a novel feed supplement to enhance protein use efficiency by reducing methane and nitrogen emission in ruminants.

## Figures and Tables

**Figure 1 molecules-27-05153-f001:**
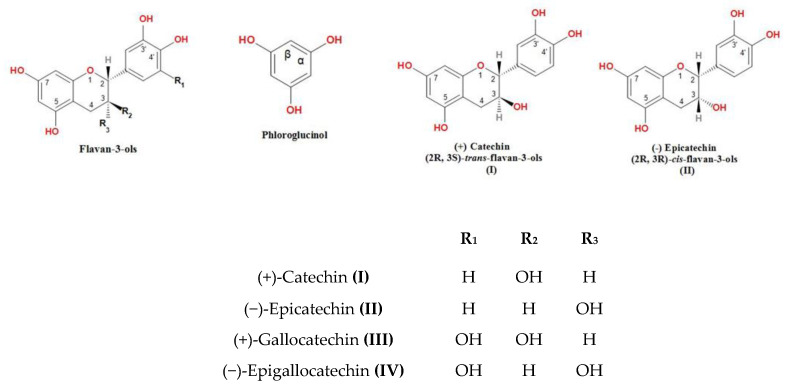
Chemical structure of the compounds investigated in the study.

**Figure 2 molecules-27-05153-f002:**
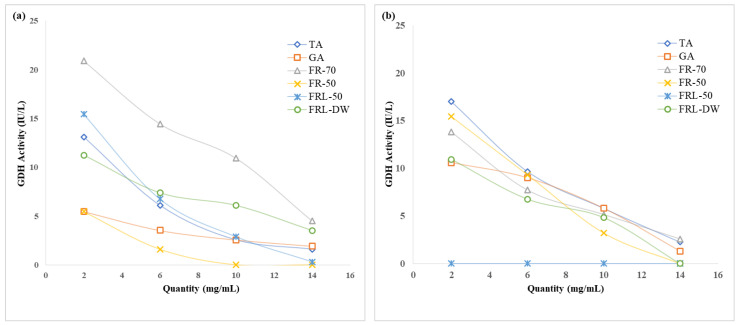
The effect of different proanthocyanin extracts. TA: Tannic acid, GA: Gallic acid, FRL-50: Extract eluted at 50% of methanol from *Ficus religiosa*, FRL-DW: Extract eluted with distilled water from *F. religiosa*, FR-70: Extract eluted in 70% acetone from *F. racemosa*, FR-50: Extract eluted in 50% methanol from *F. racemosa* with different concentrations on ruminal glutamate dehydrogenase activity (**a**) protozoal and (**b**) bacterial.

**Figure 3 molecules-27-05153-f003:**
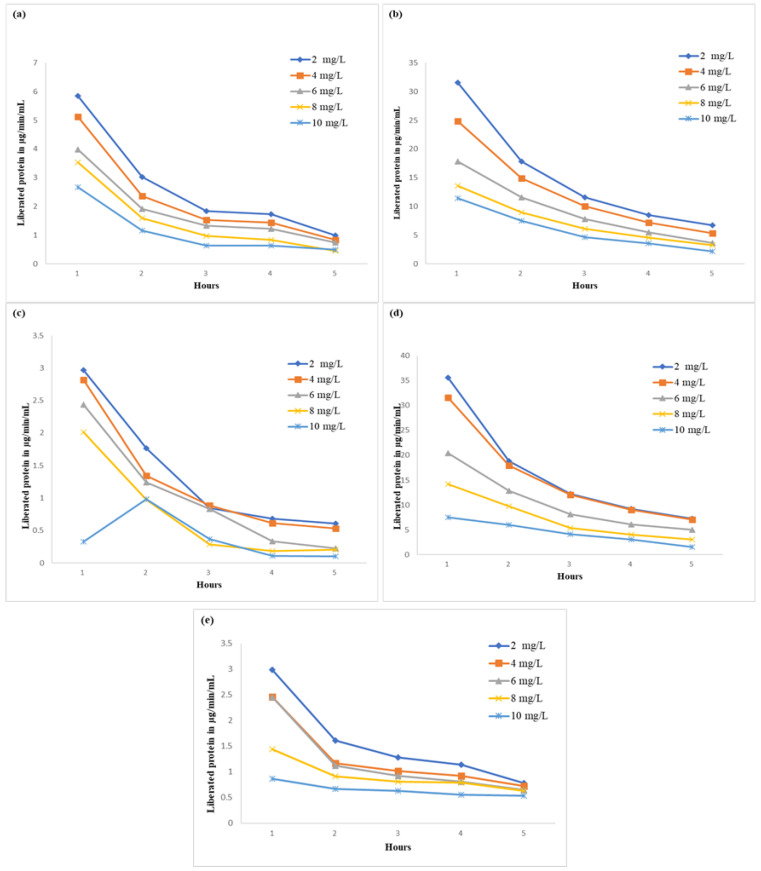
The effect of individual proanthocyanin extract on R-protease with different concentrations and times (**a**) FRL-50: Extract eluted in 50% methanol from *Ficus religiosa*, (**b**) FRL-DW: Extract eluted with distilled water from *F. religiosa*, (**c**) FR-50: Extract eluted in 50% methanol from *F. racemosa*, (**d**) FR-70: Extract eluted in 70% acetone from *F. racemosa* and (**e**) FR-DW: Extract eluted with distilled water from *F. racemosa*.

**Figure 4 molecules-27-05153-f004:**
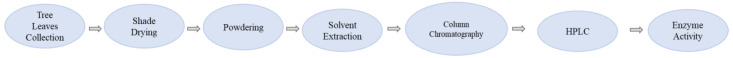
Main operational steps of plant extraction to enzyme activity.

**Table 1 molecules-27-05153-t001:** Composition of oligomers of proanthocyanidins in *F*. *racemosa* and *F. religiosa* leaves.

Components (mg/g)	FRL-50 *	FRL-DW	FR-70	FR-50	FR-DW
(+) Catechin	0.151 ± 0.01 ^b^	N.D.	0.098 ± 0.01 ^c^	N.D.	0.524 ± 0.03 ^a^
(−) Epicatechin	0.317 ± 0.02 ^a^	0.181 ± 0.03 ^bc^	0.135 ± 0.04 ^c^	0.235 ± 0.04 ^ab^	0.221 ± 0.02 ^bc^
(+) Gallocatechin	0.092 ± 0.01 ^cd^	0.249 ± 0.06 ^bc^	0.458 ± 0.13 ^a^	0.347 ± 0.05 ^ab^	0.035 ± 0.02 ^d^
(−) Epigallocatechin	0.117 ± 0.09 ^ab^	0.056 ± 0.05 ^bd^	0.065 ± 0.01 ^c^	0.016 ± 0.07 ^cd^	0.315 ± 0.11 ^a^
(+) Catechin-4-phloroglucinol	0.963 ± 0.07 ^b^	N.D.	1.461 ± 0.12 ^a^	0.990 ± 0.22 ^b^	1.361 ± 0.07 ^b^
(+) Gallocatechin-4-phloroglucinol	0.649 ± 0.09 ^b^	0.873 ± 0.11 ^b^	1.321 ± 0.17 ^a^	0.550 ± 0.09 ^c^	0.555 ± 0.10 ^c^
(−) Epigallocatechin-4-phloroglucinol	0.458 ± 0.04 ^b^	0.168 ± 0.05 ^c^	0.974 ± 0.12 ^a^	1.130 ± 0.10 ^a^	0.947 ± 0.08 ^a^
Total monomer	0.676	0.51	0.745	0.773	1.093
Total adduct	2.066	2.03	3.789	2.678	2.875
Degree of polymerization	3.06	3.99	5.09	3.46	2.63
Subunits ratio (Dp:Cy)	50:50	50:50	50:50	50:50	50:50

* *F. racemosa*, FR; *F. religiosa*, FRL. Dp: delphinidin, Cy: cyanidin, FRL-50: Extract eluted at 50% of methanol from *Ficus religiosa,* FRL-DW: Extract eluted with distilled water from *F*. *religiosa*, FR-70: Extract eluted at 70% of acetone from *F. racemosa*, FR-50: Extract eluted at 50% of methanol from *F. racemosa,* FR-DW: Extract eluted with distilled water from *F. racemosa.* N.D.: Non-detectable. Values are indicated as mean ± sd, and values within the same row with different superscripts letters are significantly different (*p* < 0.05).

**Table 2 molecules-27-05153-t002:** Effect of proanthocyanidins from *Ficus religiosa* and *F. racemosa* leaves on inhibition activity (EC_50_) of ruminal glutamic oxaloacetic transaminase (R-GOT), ruminal glutamic pyruvic transaminase (R-GPT), and ruminal R-Cellulase enzymes.

Enzymes (EC_50_ mg/mL)		TA	GA	FRL-50	FRL-DW	FR-70	FR-50
R-GOT	P	20.54 ± 1.13 ^a^	21.77 ± 0.80 ^a^	4.72 ± 0.42 ^d^	10.67 ± 1.12 ^c^	17.17 ± 1.01 ^b^	12.33 ± 1.26 ^c^
	B	17.54 ± 0.13 ^bc^	22.78 ± 0.80 ^a^	7.12 ± 0.02 ^d^	7.69 ± 1.25 ^d^	19.17 ± 1.00 ^b^	16.33 ± 0.50 ^c^
R-GPT	P	17.52 ± 0.70 ^a^	6.89 ± 0.12 ^cd^	5.44 ± 0.49 ^d^	13.59 ± 1.42 ^b^	16.42 ± 1.07 ^a^	8.61 ± 0.96 ^c^
	B	17.12 ± 0.15 ^a^	15.52 ± 0.53 ^a^	5.28 ± 0.63 ^b^	16.69 ± 1.44 ^a^	15.35 ± 0.82 ^a^	5.31 ± 0.84 ^b^
R-Cellulase	P + B	124.32 ± 1.63 ^a^	6.62 ± 0.56 ^d^	N.O.	N.O.	24.72 ± 1.25 ^b^	9.70 ± 0.54 ^c^

P: Protozoa, B: Bacteria, TA: Tannic acid, GA: Gallic acid, FRL-50: Extract eluted at 50% of methanol from *Ficus religiosa,* FRL-DW: Extract eluted with distilled water from *F*. *religiosa*, FR-70: Extract eluted at 70% of acetone from *F. racemosa*, FR-50: Extract eluted at 50% of methanol from *F. racemosa,* R-GOT: Ruminal glutamic oxaloacetic transaminase, R-GPT: Ruminal pyruvic transaminase, R-cellulase: Ruminal cellulase. N.O.: Not observed. Values are indicated as mean ± sd, and values within the same row with different superscripts letters are significantly different (*p* < 0.05).

**Table 3 molecules-27-05153-t003:** Level of microbial nitrogen and RNA quotient (NRNAQ) and % solubilization of Rubisco in ruminal substrate after in vitro digestion of *Ficus* leaves with wheat straw (WS) and berseem hay (BH).

Diet		Solubilization (%)	NRNAQ
FRL		22.36 ± 1.24 ^bc^	0.21 ± 0.02 ^bcd^
FR		17.26 ± 0.61 ^e^	0.22 ± 0.01 ^ab^
WS		8.95 ± 0.95 ^gh^	0.16 ± 0.01 ^ef^
BH		3.04 ± 0.08 ^i^	0.17 ± 0.01 ^cdef^
FR + WS	1:1	16.61 ± 1.04 ^e^	0.15 ± 0.01 ^f^
	2:1	28.58 ± 1.48 ^a^	0.22 ± 0.01 ^ab^
	3:1	15.29 ± 0.61 ^ef^	0.19 ± 0.01 ^bcde^
FR + BH	1:1	18.35 ± 1.48 ^de^	0.26 ± 0.03 ^a^
	2:1	21.70 ± 3.00 ^cd^	0.21 ± 0.01 ^bc^
	3:1	11.81 ± 1.01 ^fg^	0.21 ± 0.02 ^bc^
FRL + WS	1:1	27.95 ± 1.95 ^a^	0.19 ± 0.01 ^bcdef^
	2:1	22.47 ± 1.37 ^bc^	0.16 ± 0.01 ^ef^
	3:1	5.59 ± 0.34 ^hi^	0.18 ± 0.01 ^cdef^
FRL + BH	1:1	14.47 ± 0.74 ^ef^	0.22 ± 0.01 ^ab^
	2:1	17.80 ± 1.26 ^de^	0.17 ± 0.01 ^def^
	3:1	25.90 ± 1.18 ^ab^	0.17 ± 0.0 ^ef^

FR, *Ficus racemosa*; FRL, *F. religiosa*; WS, Wheat straw; BH, Berseem hay; NRNAQ, Nitrogen RNA Quotient. Values are indicated as mean ± sd, and values within the same column with different superscripts letters are significantly different (*p* < 0.05).

## Data Availability

All data are contained within this article and in supplementary files.

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
