# Peer review of "Sequential Extraction of Proanthocyanidin Fractions from *Ficus* Species and Their Effects on Rumen Enzyme Activities In Vitro"

_molecules, 2022, doi:10.3390/molecules27165153_

Round 1
Reviewer 1 Report
The comments are as follows:
1. The overall conclusion is missing in the abstract section.
2. The authors are encouraged to express more the importance of the study in the introduction sections.
3. More details about plant material preparation (particle size, moisture content) are necessary.
4. Have the authors tried to apply a green extraction technique for the islolation of PAs?
5. Please, improve the conclusion section with more key-findings and future perspective.
6. The authors are encouraged to cite recent publications.
7. Please, revise the text for some misprints.
Reviewer 2 Report
In this work, three proanthocyanidin fractions were sequentially extracted from plant samples.
Extraction efficiency was tested based on using different extractants with different polarity.
By this approach, the effect on rumen enzyme activities was studied in vitro.
The experimental and results parts are exhaustive and clearly presented. The English is acceptable.
REMARKS
1. In the introduction (4th paragraph), I miss a general statement about the current state as for solvent extraction that is still in use today, as this work is mainly about the solvent extraction. Follow the down below upgrade.
“Solvent extraction is still an approach-of-choice for many operators due to its reliability, well-established protocols, versatility and availability for automation [https://doi.org/10.1016/j.jchromb.2018.06.037]. This is also stand for solvent extraction of PAs [https://doi.org/10.1016/j.indcrop.2019.112040, https://doi.org/10.3389/fbioe.2022.897185]. Regarding, the most common separation techniques for PAs are column chromatography, paper chromatography, thin layer chromatography (TLC) and high-performance liquid chromatography (HPLC) [7].
2. In experimental, provide a figure of operational pipeline including sampling, sample preparation/clean up and analysis part.
3. In conclusion, provide author´s future aims regarding this scope of investigation.
Reviewer 3 Report
This is a very interesting study that evaluated several fractions of proanthocyanidin. The manuscript was well written with well-presented results.
Minor comments:
- In the abstract (line 12), the fraction ‘FR-50’ is written two times
- The solvents of the fractions must be presented in the abstract
- In addition, it is not clear what the fractions stand for. Only in the legend of table 1 is the description of each fraction, but it must be presented at the beginning of results section
- It is not clear why the authors did not find phenolic compounds in FRL-70
- Why the figure 1 did not include the (+)-gallocatechin and (-)-epigallocathechin structres?
- In Table 1, I recommend to replace the letter ‘a’ in the superscript of ‘FRL-50’ for an asterisk, because the letters are used to represent significant differences in the results.
- Why ‘FR-DW’ was not included in table 2?
- The authors should explain why they evaluated the ruminal enzymatic activities in protozoan and bacterial fractions. This explanation could sustain the differences reported in your results
Round 2
Reviewer 2 Report
Authors should include all suggested papers to cite.
Even not all seal with solvent extraction from plant.
All are important for solvent extraction development done in the recent time.
Author Response
Dear Reviewer,
Thanks for your advice and suggestion to improve the quality of our manuscript. We incorporated all three references that you suggested in support of using solvents and their improvisation with recently developed techniques in phytochemical extraction.
Hopefully, I have addressed your comments and advice.
With kind regards,
Pushpendra Koli